# Assessing landscape aesthetic values: Do clouds in photographs influence people's preferences?

**Erich Tasser**[1], **Alexandros A. Lavdas**[2,3], **Uta Schirpke**[1,4,5]*

**1** Institute for Alpine Environment, Eurac Research, Bozen/Bolzano, Italy, **2** Institute for Biomedicine, Affiliated Institute of the University of Lübeck, Eurac Research, Bozen/Bolzano, Italy, **3** The Human Architecture & Planning Institute, Inc., Concord, MA, United States of America, **4** Department of Ecology, University of Innsbruck, Innsbruck, Austria, **5** Department of Geography, Ludwig-Maximilian-University, Munich, Germany

* Uta.Schirpke@eurac.edu

**Data Availability Statement:** All relevant data are within the paper and its Supporting Information files.

**Funding:** This work was supported by the Department of Innovation, Research, University

## Abstract

Photo-based surveys are widely applied to elicit landscape preferences and to assess cultural ecosystem services. Variations in weather and light conditions can potentially alter people's preferences, as sunny landscapes are more positively perceived than those under inclement weather conditions. To assure comparability across pictures, studies usually include photographs taken at sunny days (i.e., blue sky). However, the influence of clouds in sunny landscapes on people's preferences has been rarely considered, although color contrasts between clouds and the blue sky may attract people's attention. This study therefore aimed to assess the effects of clouds in landscape photographs on people's preferences by (1) examining differences in preference between pairs of landscape photographs (i.e., with clouds and without clouds), and (2) explaining variations through variables from eye-tracking simulation, photo content analysis, and Geographic Information System (GIS)-based analysis. Our results indicate no significant differences in preferences between pictures with and without clouds when the pictures with clouds contained a proportion of sky around 22% and a cloud cover of about 39%. However, a higher proportion of sky positively influenced landscape preferences, while a higher proportion of clouds, especially in combination with a lower proportion of sky, had negative effects. These findings suggest that landscape preference studies should pay attention not only to the appearance of the sky in terms of cloudiness, but they also should control the proportion of sky across different pictures to obtain comparable results. Future research should address limitations regarding the transferability of our findings to other types of landscapes and regarding potential differences in perceptions between respondents with different socio-cultural characteristics. Moreover, landscape preferences under changing weather conditions or different cloud types as well as diurnal and seasonal changes should be further explored.

and Museums of the Autonomous Province of Bozen/Bolzano. The authors thank the Department of Innovation, Research, University and Museums of the Autonomous Province of Bozen/Bolzano for covering the Open Access publication costs. The funders had no role in study design, data collection and analysis, decision to publish, or preparation of the manuscript.

**Competing interests:** The authors have declared that no competing interests exist.

## Introduction

Landscapes provide a wide range of cultural ecosystem services such as recreational, aesthetic, educational, and spiritual values [1]. Aesthetic values are one of the most appreciated cultural ecosystem services [2–4], associated with a wide range of subjective benefits to people such as spirituality, knowledge, identity, and health [5]. Also, many aspects of objective physical and psychological health benefits of exposure to natural landscapes have been well documented, from stress reduction to improved recovery from surgery [6–8]. Appealing landscapes also can provide economic benefits [9], as they are strongly linked to outdoor recreation [10, 11]. Furthermore, high aesthetic quality of landscapes can support biodiversity conservation efforts [12, 13]. Due to rapid landscape transformations caused by global change pressures, decision-makers need reliable information on people's landscape preferences to develop sustainable management strategies and, at the same time, to preserve or restore attractive landscapes [14, 15].

Aesthetic landscape values arise from the interaction of people with the biophysical characteristics of the natural environment [16], and are usually assessed applying two distinct approaches. While expert-based approaches analyze visual landscape properties through quantitative information such as indicators describing landscape patterns [17, 18], perception-based approaches consider subjective values associated to the landscape [19]. Despite the subjectivity of people's perceptions, perception-based approaches are considered to provide reliable results for the general public [20]. Nevertheless, landscape preferences of individuals may be strongly heterogeneous [21] and are often influenced by socio-cultural characteristics [17, 22–27]. In perception-based approaches, questionnaires eliciting landscape preferences on the basis of photographs are the most frequently used methodology [22–25, 28, 29]. These studies usually ask the participants to evaluate the photographs using rating scales, e.g., ranging from 1–5 [17, 27], 1–6 [30], or 1–10 [31, 32], or applying discrete choice experiments [23, 33, 34]. Photographs are considered reliable stimuli to gather people's preferences, comparable to on-site surveys [35]. A big advantage of using photographs is that it allows researchers to involve a high number of respondents, which enables the generation of a large database and a strong data analysis [36]. Moreover, landscape pictures can be manipulated or digitally designed to control the image content for assessing specific aspects [37–39].

To guarantee comparability across different photographs, studies usually use pictures taken at sunny days, i.e., blue sky [24, 25, 30]. Weather conditions in the photographs can alter people's preferences, as the degree of cloudiness and the proportion of cloud cover in the sky affect the illumination of the landscape, i.e., the landscape appears darker with less contrast and details and colors are less discernable compared to clear weather conditions [40]. Thus, inclement weather conditions can moderate preferences for waterscapes [41] and associated benefits such as happiness [42]. In contrast, sunrise, rainbows, and sunset can significantly increase preferences compared to blue sky [43]. In arts and photography, which have been intensively concerned with the aesthetics and emotional effects of appearance of the sky, clouds are often used to convey certain moods and messages [44, 45]. For example, since the sky symbolized the supernatural instance, light and rays were considered symbols of the divine, thunderclouds and lightning as symbols of God's punishment [46]. The sky and cloud patterns were also used by artists to depict their emotions and inner states, which gave rise to "mood painting", where the weather in combination with the landscape was used as a symbol or allegory for feelings [44, 47]. The importance of clouds, also as mood-setters, was underpinned by the introduction of the term cloudscapes [44]. The appearance of different cloud types ranges from small to big puffy shapes (e.g., low-altitude cumulus or nimbostratus clouds), to wavy shape (e.g., mid-level altitude altocumulus clouds) and patchy or wispy shapes (e.g., high altitude cirrus, cirrostratus) [48]. Landscape photography also took up the subject of

clouds to influence the aesthetics and emotional impact of photographs [49]. Accordingly, photographs posted in social media often reflect weather phenomena capturing the atmosphere (mood) of a place and aesthetic experiences of landscapes [50, 51].

For these reasons, perception-based surveys, using photographs to gather landscape preferences, usually include only pictures taken at sunny days. However, it remains unclear how the presence of clouds in those photographs also alters preferences. We hypothesize that clouds in landscape pictures may influence the stated preferences due to color contrasts between clouds and the blue sky, as color contrasts, hue variations, and edges attract people's attention when looking at landscape pictures [29, 52, 53]. Moreover, the preferences may be influenced by emotions and moods that clouds can bring to a landscape [45, 49, 51]. Thus, this study aimed to assess the effect of clouds in landscape photographs on people's preferences. The analysis is guided by two research questions: (R1) Is there a difference in preference between a landscape photograph with clouds and without clouds? (R2) If differences in preference occur, how can these variations be explained? Our findings provide valuable information for designing and evaluating photo-based assessments in landscape research.

## Materials and methods

### Study design

To answer the two guiding research questions R1 and R2, we applied different analysis steps based on landscape photographs (Fig 1, see following sections). The original photographs (A) had different degrees of cloud cover, and a cloud-free version (B) was created for each photograph. People's preferences were collected in an online survey. Predictor variables were derived from three methodological approaches, including eye-tracking simulation, photo content analysis, and Geographic Information System (GIS)-based analysis [53]. Finally, statistical analyses were carried out to test for differences between picture pairs (R1) and to explain variations in preferences scores (R2).

### Photo-based survey

To evaluate the influence of clouds on landscape preferences, we used landscape photographs from two previous studies [25, 54]. The photographs were taken in various locations of the Central European Alps and depicted different mountain landscapes that were typical for this area, including rural settlement areas, agricultural areas, alpine pastures, and high mountain landscapes. All photographs showed a 360° panorama and were taken at normal eye level (approx. 1.5–1.7 m) during sunny summer days. When taking the pictures, attention was paid that the horizon did not extend over the center of the picture [55]. Despite a blue sky, the photographs contained variations in cloud patterns, ranging from almost no clouds to a sky largely covered by clouds, but no threatening cloud atmosphere.

From this pool of 49 landscape photographs [25, 54], we selected 29 photographs with varying cloud cover (S1 Fig) to not overload the respondents with a too high number of pictures. The sky of all original photographs (A) was manipulated by removing the clouds and recoloring these areas to derive a version of the photograph without clouds (B), using Adobe Photoshop image processing software (Adobe Photoshop CS6, Adobe Systems Inc, CA, USA). The manipulation only altered the appearance of the sky, and no landscape features were changed (S1 Fig).

All original (A) and manipulated (B) photographs (n = 58) were arranged in an online questionnaire using LimeSurvey (LimeSurvey GmbH, Hamburg, Germany; www.limesurvey.org/). The questionnaire was prepared in German and contained three sections. In the first section, participants were informed about the scope of the survey and that the participation in the survey was voluntary and anonymous (see ethics statement below). In the second section, we

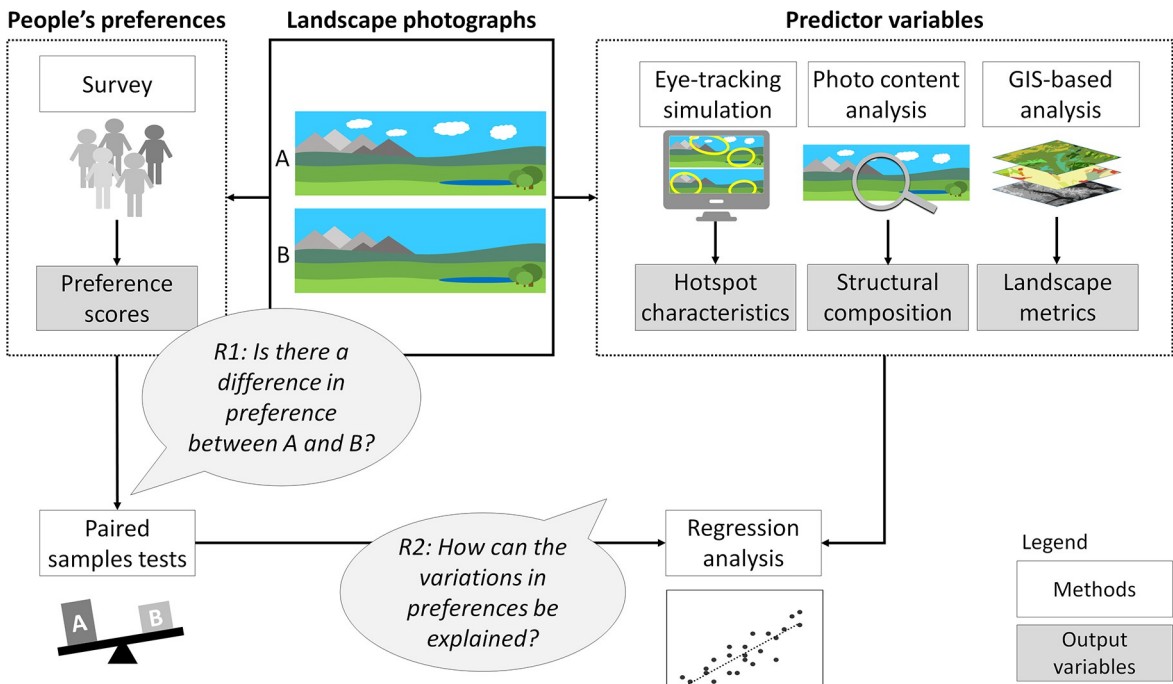

**Fig 1. Analysis steps to assess the effect of clouds in landscape photographs on people's preferences.** For each photograph (A), a manipulated version without clouds was created (B). People's preferences were collected in an online survey and the picture pairs were tested for differences in preferences between A and B to answer research question R1. For all pictures, various predictor variables were calculated using three different approaches [53]. The variables were related to the preference scores via regression analysis to explain variations in preference scores and answer research question R2.

asked the respondents to express their preference for each photograph on a 10-point Likert scale (1 = 'I don't like it at all' to 10 = 'I like it very much'). There is no consensus among landscape preference studies on the most appropriate rating scale. We used an even scale, because we wanted the respondents to decide whether their preference for an image is more positive or negative. Moreover, response scales from 7 to 10 are most reliable, and respondents prefer the 10-point scale the most [56]. All participants assessed all 58 photographs, but the order of the pictures was randomly arranged for each respondent to avoid priming effects [57]. In the third section, participants could also indicate socio-demographic information such as gender, age group, language group, place of residence (city, country), school education and field of work.

The online survey was carried out between March and July 2022, using a convenience sampling approach. We invited students in ecology of the University of Innsbruck (Austria) and students in physical geography of the Ludwig Maximilian University in Munich (Germany) during lectures on applied ecology and on environmental management to participate in the survey by providing a link to the questionnaire. We also asked them to forward the link to friends and relatives (particularly parents/grandparents, but no minors). The participants received neither a compensation nor other incentives. They completed the questionnaire in an unsupervised setting using a digital device and the responses were saved in a database. We estimated that the time to fill it out was about 10 minutes. There were no time restrictions in viewing and rating the photographs. The participants could also pause the survey by saving their responses and continue at a later time.

To ensure a minimum sample size for statistical analysis, i.e., to set up a significant regression model, we determined the minimum sample size through statistical power analysis [58].

With a determination coefficient of $R^2 = 0.7$, a statistical power of 0.9 and a significance level of $\alpha = 0.01$, we needed a sample size of at least 68 valid responses. In total, 176 people opened the questionnaire and 124 participants completed it. The participants were mostly female (70.3%), under 25 years old (66.4%), and residents in North Tyrol (69.2%). Most of them had a highschool degree (69.7%) and were not (yet) employed/still in eduction (63.3%). For full demographic details, see S1 Table.

Due to the convenience sampling approach, we first evaluated the representativeness of the sample, i.e., whether our results were specific for this group of respondents or whether they matched the preferences of the general public. For all pictures, we first calculated mean preference scores and then compared those of the original photographs in our study to the results of the two former surveys, from which we selected the photographs [25, 54]. Both studies gathered landscape preferences in a comparable way between 2016 and 2019 in five different study areas in the Central Alps. Both studies used a stratified sampling approach based on the demographic distribution in the study areas, i.e., considering key socio-demographic characteristics such as gender, age, origin (residents, tourists), place of living (village, city), and native language (German, Italian). The two surveys, comprising 967 respondents [25] and 384 respondents [54], can therefore be considered to be representative for the general public of the Central Alps. In both surveys, differences between socio-demographic groups in preference scores were generally very small, although some significant differences occurred, mostly between people of different language, age, and origin [25]. After aligning the preference scores among the three surveys using the linear stretch method [59], we calculated the degree of correlation applying linear regression analysis. Since we obtained a high correlation ($R^2 = 0.842$; $p<0.001$; S2 Fig), we can assume that our sample also represents well enough general preferences.

**Ethics statement.** The participants were informed before filling out the questionnaire that participation in the survey was voluntary and anonymous and that no personal data would be requested that would allow the person to be identified. Access to the questionnaire was only granted if the person actively requested it and gave his/her the consent by checking the box before starting the questionnaire. The information for this study was collected in a manner, in which the institutions did not collect personal data that would allow identification of individual human participants. According to the institutional and national rules, this study did not need approval of an ethics committee, as it was not a clinical study treating sensitive personal data nor health-related information. Only adults were invited to participate. The study was in line with the principles established by national and international regulations, including the Declaration of Helsinki (64th WMA General Assembly, Fortaleza, Brazil, Oct 2013) and the Code of Ethics.

## Collection of variables

To explain variations in preferences, we prepared a set of predictor variables for each picture (Tables 1 and S2–S4), using three different methodological approaches [53] as described in the following.

**(1) Eye-tracking simulation.** Eye-tracking simulation software 3M-VAS (3M™ Visual Attention Software, St. Paul/Minnesota, USA; https://vas.3m.com) was used to identify hotspots of initial eye-tracking movements on the pictures. 3M-VAS uses an artificial intelligence (AI) algorithm that is based on 30 years of lab-based eye-tracking research to simulate human pre-attentive processing in vision [60]. Based on five visual elements that are recognized to attract human attention (edges, intensity, red-green color contrast, blue-yellow color contrast, and faces), the algorithm predicts what people would be focusing their eyes on before the

**Table 1. Variables derived for all pictures using three different analysis approaches [53].**

| Analysis approach | Variables | Unit | Description |
|---|---|---|---|
| Eye-tracking simulation | Hotspot area | % | Total estimated area of the hotspots in the photo |
| | Top-hotspot area | % | Estimated area of the top-hotspot |
| | Lowest hotspot probability | % | Probability of initial eye-tracking movement of the hotspot with the lowest probability |
| | Sky within hotspots | % | Estimated area of sky horizon within the hotspots |
| | Intensity of visual elements | % | Estimated mean contribution of intensity of visual elements (i.e., luminance contrast, brightness, black/white contrast) within the hotspots to the overall probability of the hotspot |
| | Red-green color contrast | % | Estimated mean contribution of red-green color contrast of visual elements within the hotspots to the overall probability of the hotspot |
| | Blue-yellow color contrast | % | Estimated mean contribution of blue-yellow color contrast of visual elements within the hotspots to the overall probability of the hotspot |
| Photo content analysis | Sky | % | Estimated area of sky within the photo |
| | Clouds | % | Estimated area of clouds within the sky |
| | Artificial elements | % | Estimated area of clearly recognizable artificial elements within the photo (e.g., street, street signs, cars, fences) |
| | Percentage near zone | % | Estimated near zone (0–60 m) within the photo |
| | Percentage middle zone | % | Estimated middle zone (0.06–1.5 km) within the photo |
| | Percentage far zone | % | Estimated far zone (>1.5 km) within the photo |
| GIS-based analysis | Patch density | n ha$^{-1}$ | Patch density |
| | Largest patch index | index | Largest patch index |
| | Area-weighted mean of patch area | ha | Area-weighted mean patch area distribution |
| | Area near zone | ha | Total visible area of near zone (<1.5 km) |
| | Area middle zone | ha | Total visible area of middle zone (1.5–10 km) |

conscious brain can react, i.e., in the initial 3–5 seconds of looking at an image [60]. 3M-VAS claims to be as reliable and effective as eye tracking hardware, predicting pre-attentive vision with around 92% accuracy [60]. The software also overcomes issues related to repeatability of real eye tracking experiments and allows direct comparison of the outputs [61].

To obtain the simulated eye-tracking results, we uploaded the pictures to the 3M-VAS website, which were then processed under a minute. The pictures were standardized size (2300x360 pixel) and resolution (300 dpi), as used in the questionnaires. We scanned all pictures using the category "Other," representing the most general and unbiased modality [62]. The analysis report of 3M-VAS included (1) a heatmap, (2) the hotspots that were derived from the heatmap, specifying the probability that a person will look somewhere within the hotspot areas within the first 3–5 seconds, (3) a gaze sequence indicating the most probable viewing order of the 4 most-likely seen hotspots, and (4) a report of visual elements (i.e., edges, intensity, red-green color contrast, blue-yellow color contrast, faces) indicating their contribution to the overall probability, from which the heatmap was derived [60]. All output images had a size of 1024x160 pixel with a resolution of 96 dpi. As 3M-VAS analyzes each picture individually, the results are independent and comparable [61]. To estimate different quantitative information for each pictures, we used the hotspots as well as the report on the visual elements (Tables 1 and S2). From the hotspots (S3 Fig), we derived, for example, the area of the hotspots, probabilities of initially viewing the hotspots, and the area of sky within the hotspots. To systematically analyze all hotspots, we overlaid a grid with 3x9 cells over the 3M-VAS output (S4 Fig). In each grid cell, we counted the coverage (%) of the hotspots and summarized the percentages across all cells. Furthermore, the contribution of the visual elements (edges, intensity, red-green color contrast, blue-yellow color contrast, faces) was estimated for the hotspots.

For this purpose, we estimated the spatial coverage of the edges within each hotspot from 0 to 100% (S5 Fig). For the other elements, we estimated the level of the color intensity, ranging from no value (black = 0%) to high values (white = 100%) (S5 Fig). Finally, the values for each visual element were averaged across all hotspots to obtain their contribution to the probability of the hotspots in each photograph. All estimations were carried out by one researcher to assure comparability of the estimations. For further details, see Schirpke et al. [53].

(2) **Photo content analysis.**   In the photo content analysis, we estimated the structural composition of the pictures (S3 Table) by visual analysis, including the percentage of sky and cloud cover. Using the same grid with 3x9 cells (S4 Fig), the area of landscape features was estimated by one researcher counting the coverage (%) of the features in each cell and summarizing the percentages across all cells. The same person also estimated the composition of the landscape in terms of distance zones (i.e., near zone, middle zone, far zone; S4 Fig). In the near zone (0–60 m), individual landscape features (e.g., trees, buildings) could be clearly identified [63]. The middle zone extended to about 1.5 km from the photo location and individual landscape features were still clearly discernible. In the far zone, landscape features were not visible and only different land cover types such as grassland, forest, settlement areas were distinguishable.

(3) **GIS-based analysis.**   GIS-based analysis was applied to quantify the composition and configuration of the landscape that was depicted on the photographs through landscape metrics [64]. In a first step, viewsheds up to 50 km [65] were calculated from the photo locations based on a digital surface model (DSM). For each cell of the DSM, viewshed analysis determines whether the cell is within the observer's line-of-sight or not [66], i.e., which areas on the map are visible and which areas are hidden by vegetation, buildings, or mountains. In a second step, the visible area was overlaid with land-use/cover maps, while non-visible areas were excluded from further analysis. In this way, we aligned the content of the map with the content of the photographs. To account for changes in discernibility of landscape features (see above), we used different habitat and land cover maps as well as different spatial resolution for different distance zones (S4 Table). Finally, we composed the resulting maps of the different zones into one single map. Spatial analysis and map preparation was done in ArcGIS 10.4™ (ESRI, Redlands, CA, USA). The final map was used to calculate 12 landscape metrics (S5 Table) with FRAGSTATS Version 4.2 [64]. For full details, see Schirpke et al. [67].

## Cloud types and spatial arrangement

In addition to the predictor variables, we determined the cloud type of each original photograph. Furthermore, the photo composition in terms of spatial arrangement, e.g., position of horizon, can influence people's preferences [55, 68]. Based on the Rule of Thirds, as a simplification of the golden section or golden ratio (Phi, ϕ), we divided the picture into thirds both horizontally and vertically [69, 70]. This grid with 3x3 cells (S6 Fig) was overlaid over the hotspots to determine a spatial shift between pictures for which the preference scores significantly differed between the original picture with clouds and the manipulated picture without clouds.

## Statistical analyses

To assess differences in the preference scores between the original landscape photograph with clouds (A) and the manipulated picture without clouds (B), we performed statistical comparisons using a bootstrap paired samples t-tests with 10,000 iterations. Significant differences were set as $p < 0.05$. We calculated mean values of the variables sky and clouds across picture pairs with decreased (A>B), not changed (A = B), and increased (A<B)

preference score. Statistically significant differences were tested using the least significant difference (LSD) post hoc test.

To explain variations in landscape preferences between the picture pairs, we applied backward stepwise linear regression. The backward stepwise approach starts from a full model with all predictors and successively reduces them to obtain a model that best explains the data, while solving problems of multicollinearity and overfitting [71]. We used the difference in preference score (B-A) between the manipulated picture without clouds (B) and the original landscape photograph with clouds (A) as dependent variable and all variables from the eye-tracking simulation, photo content analysis, and GIS-based analysis as independent predictor variables (S6 Table). We removed collinear variables to avoid overfitting and to support the interpretability of the model by first screening for multicollinearity among the predictor variables. We calculated the variance inflation factor (VIF) values for each variable and removed variables with a VIF <10. Then, we applied a backward stepwise linear regression routine to further diminish the number of variables [72] and iteratively excluded variables with low predicting performance until no further improvement was possible (Table 1). To assess the level of variability explained by the model, we used the difference between $R^2$ and the adjusted $R^2$, as the adjusted $R^2$ is less influenced by overfitting than $R^2$ [73]. Through analysis of variance (ANOVA) or t-tests, validity, quality, and significance of the coefficients were evaluated. The reliability of the regression was determined using the Shapiro-Wilk test (residual normality), the Durbin-Watson test (autocorrelation of the variables), the Breusch-Pagan test (homoscedasticity) and the VIF statistics (multicollinearity effect). All preconditions for the reliability of the regression using were fulfilled (Shapiro-Wilk test for residual normality: $p = 0.503$; Durbin-Watson test for autocorrelation of the variables: 2.143), Breusch-Pagan test for homoscedasticity: $p = 0.326$) and the VIF statistics (multicollinearity effect, see Table 2). All statistical analyses were performed in SPSS Statistics (IBM SPSS 27, NY, USA).

## Results

Comparing the preference scores between the original landscape photograph with clouds and the manipulated picture without clouds, landscape preferences did not significantly differ in about half of the cases (n = 15). For fourteen picture pairs, differences in preference scores were statistically significant. In nine cases, preference scores were lower with clouds, while the removal of clouds increased the preference scores in five cases (Fig 2 and S7 Table).

Comparing the mean proportion of sky and clouds of the picture pairs with decreased (A>B), not changed (A = B) and increased (A<B) preference score revealed opposing effects of cloud removal on preference scores (Fig 3 and S8 Table). The removal of clouds in photographs with a high proportion of sky (34%) and few clouds (18%) led to a decrease in preference score (A>B). In contrast, if the original photograph contained a low proportion of sky (16%) and a high degree of cloud cover (44%), preference scores were higher for the manipulated picture (A<B). For pictures with a proportion of sky covering around 22% of the photograph and a low level of cloud cover (about 39%), the removal of the clouds had no effect on the preference score (A = B). In addition, the hotspots were located more towards the sides than in the center in the more attractive image (S9 Table). In 10 of 14 cases, there was a significant shift of hotspots from the center to the top or bottom corners of the photos with cloud removal (pictures 3, 5, 6, 9, 11, 14, 22, 26, 28, 29). In 3 cases, hotspots did not shift (pictures 2, 13, 24), and in only one case, hotspots moved from the sides to the center (picture 20). Pictures with single, smaller clouds (mostly cumulus or cirrostratus clouds) obtained higher preference scores, while in pictures with lower preference due to a high cloud cover, nimbostratus clouds occurred more often (S10 Table).

**Table 2. Result of the multiple linear regression, using the difference in preference score (B-A) between the manipulated picture without clouds (B) and the original landscape photograph with clouds (A) as dependent variable.** Positive differences indicate that the photo without clouds is preferred over the photo with clouds, negative values indicate the opposite result. Accordingly, a negative regression coefficient B indicates that this variable has a positive effect on the preference score of photos with clouds. Only predictors with variance inflation factor (VIF) <10 during collinearity diagnostics are included.

| Variables | Non standardized coefficient | | Standardized coefficient Beta | T | Sig. | VIF |
|---|---|---|---|---|---|---|
| | Regression coefficient B | SD | | | | |
| (Constant) | 1.755 | 0.298 | | 50.893 | <0.001 | |
| Hotspot area (%) | -0.026 | 0.005 | -0.583 | -40.864 | <0.001 | 2.940 |
| Top-hotspot area (%) | 0.085 | 0.015 | 0.848 | 50.713 | <0.001 | 4.514 |
| Lowest hotspot probability (%) | -0.011 | 0.004 | -0.462 | -30.046 | 0.012 | 4.713 |
| Sky within hotspots (%) | -0.026 | 0.005 | -0.911 | -50.259 | <0.001 | 6.152 |
| Intensity of visual elements (%) | 0.005 | 0.003 | 0.274 | 20.015 | 0.072 | 3.800 |
| Red-green color contrast (%) | -0.013 | 0.002 | -0.909 | -50.421 | <0.001 | 5.757 |
| Blue-yellow color contrast (%) | 0.008 | 0.003 | 0.424 | 30.075 | 0.012 | 3.902 |
| Sky (%) | -0.029 | 0.003 | -10.593 | -100.572 | <0.001 | 4.652 |
| Clouds (%) | 0.003 | 0.001 | 0.353 | 30.051 | 0.012 | 2.750 |
| Artificial elements (%) | -0.011 | 0.002 | -0.602 | -50.177 | <0.001 | 2.771 |
| Percentage near zone (%) | 0.007 | 0.003 | 0.299 | 20.308 | 0.044 | 3.438 |
| Percentage middle zone (%) | 0.010 | 0.003 | 0.763 | 30.918 | 0.003 | 7.763 |
| Percentage far zone (%) | -0.015 | 0.005 | -0.376 | -20.970 | 0.014 | 3.277 |
| Patch density (n ha$^{-1}$) | -0.093 | 0.016 | -0.910 | -50.856 | <0.001 | 4.950 |
| Largest patch index (index) | -0.004 | 0.002 | -0.191 | -10.860 | 0.093 | 2.164 |
| Area-weighted mean of patch area (ha) | 0.008 | 0.001 | 0.838 | 60.078 | <0.001 | 3.893 |
| Area near zone (ha) | -0.001 | 0.000 | -0.293 | -20.767 | 0.020 | 2.295 |
| Area middle zone (ha) | 0.000 | 0.000 | -0.958 | -50.162 | <0.001 | 7.061 |

The difference in preferences between picture pairs could be explained by seven hotspot characteristics, six photo content variables, and six landscape metrics (Table 2). The stepwise linear regression analysis led to a model explanation of 95.1% of the differences in preferences

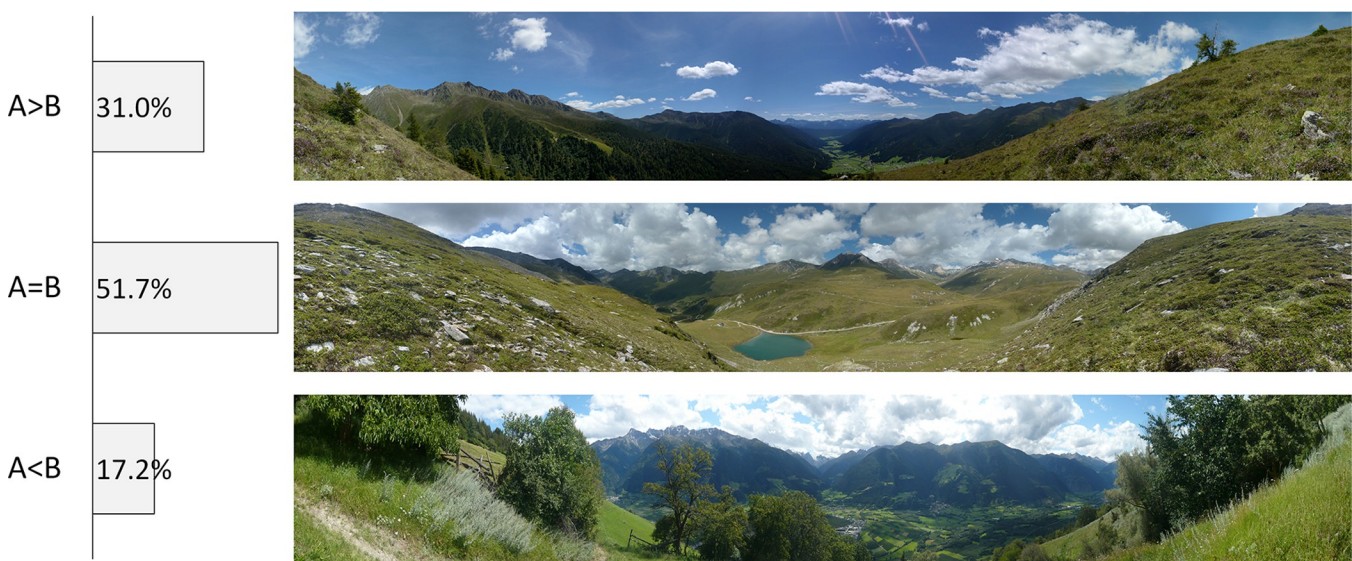

**Fig 2. Percentage of photos with significant differences in preference scores due to the removal of clouds (paired sample t-test, $p < 0.05$) and photo examples.** Photographs by Eurac Research.

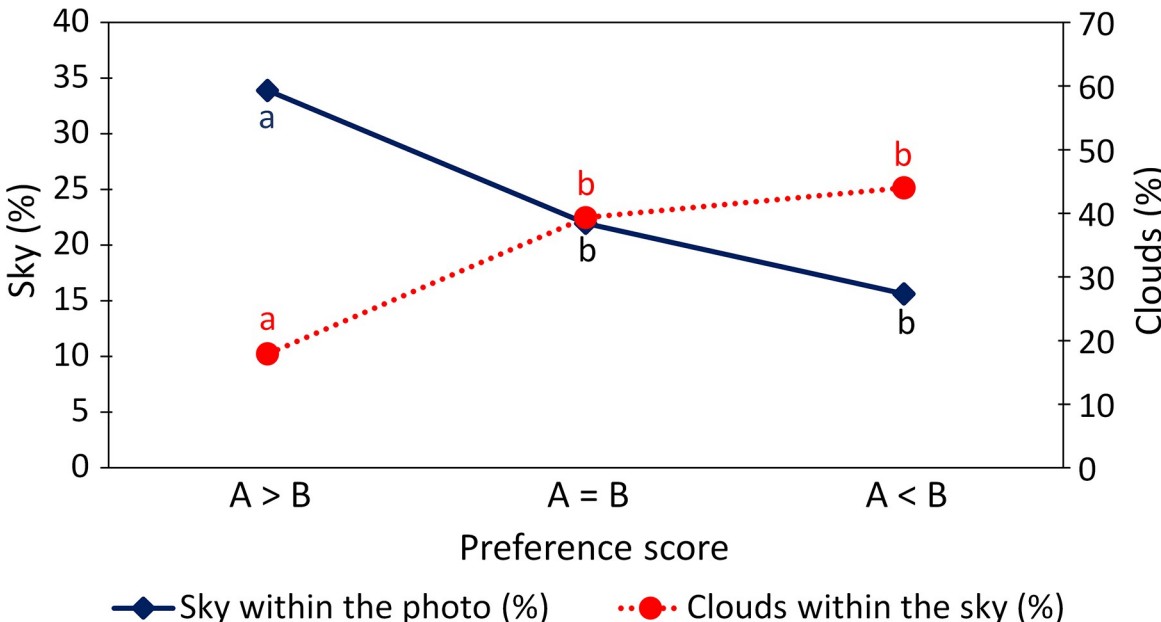

**Fig 3. Relationship between the proportion of sky and clouds in the picture and effects on the preference score due to the removal of clouds.** Values represent mean values of the variables sky and clouds across picture pairs with decreased (A>B), not changed (A = B), and increased (A<B) preference score. A: Original picture with clouds, B: manipulated picture without clouds. Different colored letters (a, b) near the symbols indicate statistically significant differences (LSD post hoc test, p < 0.05).

(ANOVA p<0.001; $R^2$ = 0.951; adjust. $R^2$ = 0.863). The variables derived from eye-tracking simulation indicated that higher rated pictures with clouds comprised on average a higher percentage of hotspot areas, also containing a part of the horizon or sky (Table 2) than the modified picture without clouds. The hotspots were often characterized by high red-green contrast (see Example 1, Fig 4). In contrast, preferences for pictures without clouds were often related to the presence of a large top hotspot, a high intensity of visual elements, and high blue-yellow contrasts (Table 2).

In terms of photo content, landscape preferences for pictures with clouds were related to a higher proportion of sky, provided that the degree of cloud cover was not too high (Table 2). In these pictures, a high proportion of background area had a positive influence on landscape preferences, while artificial elements generally negatively influenced preferences. Contrastingly, high proportions of foreground and middle-ground reduced preferences for non-cloudy landscapes (see Example 2, Fig 4).

Landscape patterns also had different influence on landscape preferences for pictures with and without clouds (Table 2). For pictures with clouds, a high patch density with a large central land cover type (Largest patch index) and a high proportion of the foreground was positively related to landscape preferences. In contrast, a more homogeneous landscape composition (Area-weighted mean of patch area) and a higher proportion of the middle ground increased landscape preferences for pictures without clouds.

## Discussion

### Perception of sky and clouds

In this study, the hypothesis that clouds in landscape photographs can influence the stated preferences due to color contrasts between clouds and the blue sky was partially confirmed.

Example 1: Landscape preference A > B (difference = -0.593)

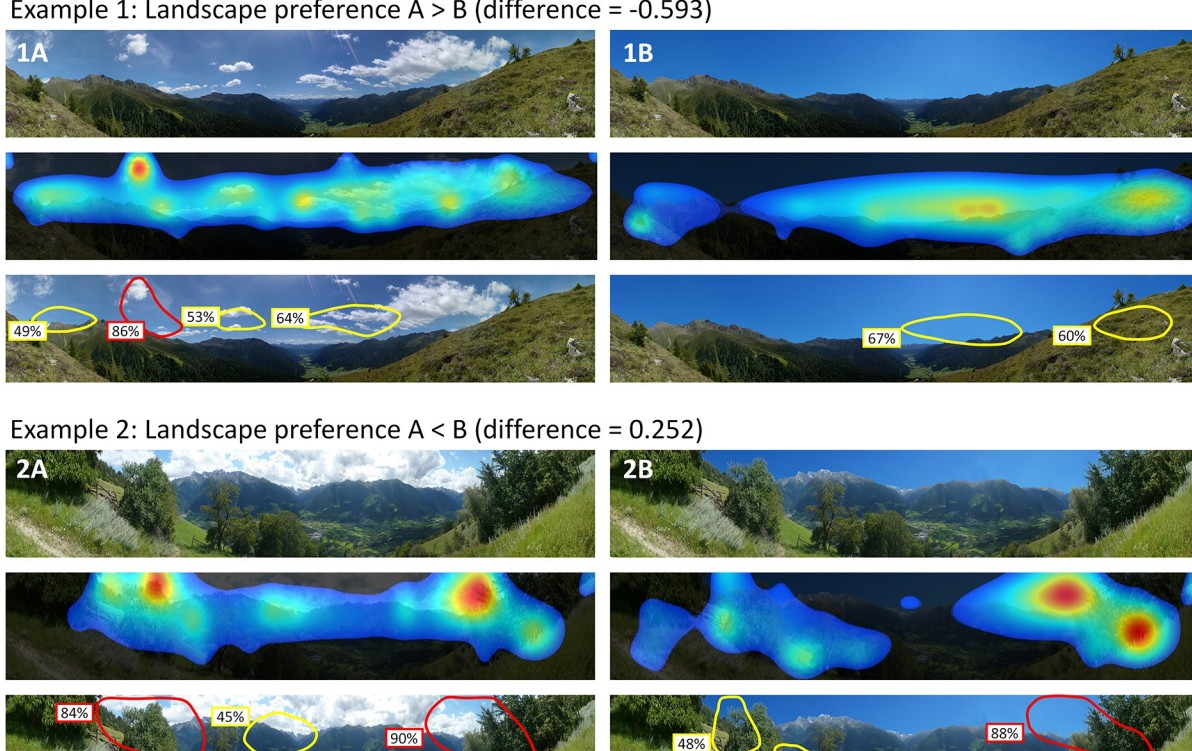

Example 2: Landscape preference A < B (difference = 0.252)

**Fig 4. Divergent effects of clouds on landscape preferences in two landscapes (Examples 1 and 2).** For each example, above are shown the original picture with clouds (A; photographs by Eurac research) and the manipulated picture without clouds (B), the heatmap indicating the probability that areas are seen within the first 3–5 seconds (middle), and the hotspots derived from the heatmap with the probability that a person will look somewhere within the hotspot areas within the first 3–5 seconds (below). Hotspots are delimited with a red line a probability >= 80% and with a yellow line for a probability < 80%.

Significant differences in preferences were dependent on the percentage of cloud cover and proportion of the sky. Our findings indicate that a high proportion of clouds leads to lower preference scores compared to a sky without clouds. In these pictures, the percentage of clouds in the sky was on average 44% and the sky sometimes appeared darker, for example, due to nimbostratus clouds. This aligns with studies showing that inclement weather conditions or ephemeral phenomena such as thunderstorms lead to lower landscape preferences [41, 43]. In contrast, our results also indicate that single, smaller clouds (e.g., cumulus clouds) positively influenced preference scores. The visibility of the fractal outline of clouds [74] in these cases, helped by its clear contrast to the sky, might be part of the explanation for this. Such a finding fits well with our current understanding of the preferential perception of geometrical structures based on a hierarchy of scales, as in the nested symmetries of a fern leaf, for example, see Taylor et al. [75] as a review. Functional magnetic resonance imaging (fMRI) work has revealed the potential neural correlates of this phenomenon, as the processing of recursive forms has been shown to recruit different resources to the processing of non-recursive forms. Remarkably, it has been shown to recruit the default mode network, a functional brain network known to be involved in the processing of internal information [76, 77], thus implying a less computation-intensive processing of such shapes. In the pictures with higher preference, the eye-tracking hotspots were also often located on such small but high-contrast clouds or

smaller cloud groups. Similarly, the study by Smalley and White [43] demonstrates that the appearance of the sky during sunrise or sunset increases perceived beauty of landscapes.

Regarding the composition of the pictures, painting and photography indicate clear rules, which can be related to our findings. The golden section or golden ratio (Phi, ϕ), referring to the positioning of the main subject in the picture or photograph, is considered the ideal ratio value determining aesthetic preferences, although controversially discussed [55, 78]. It is based on the fact that the human brain perceives images as harmonious if this rule is respected. As a simplification of the golden ratio, the Rule of Thirds is also central in design, films, and paintings, as well as in photography [69, 70]. Visual preferences have been found for photographs that depict about one third sky and two thirds foreground, if the foreground is interesting and striking [55]. In contrast, if the foreground is inconspicuous and/or there is a beautiful sky with clouds, then the rule of thirds is reversed, i.e., about one third foreground and two thirds sky [55]. Regarding our results, this means that landscape preferences may be higher if the proportion of sky reaches up to one third of the picture compared to another photograph that has considerably less proportion of sky. The vertical alignment was less important in our study, as the landscape did not change between the picture pairs. Nevertheless, the hotspots in the more attractive pair were located more towards the sides due to the lack or presence of clouds.

## Implications for landscape research

Our findings also provide insights for landscape research. Many landscape preference studies using photographs as stimuli included pictures with blue or clear sky [24, 25, 30, 79], which is still recommendable in the light of our findings. There are also no constraints to include pictures with a proportion of sky around 22% and a cloud cover of about 39%. Nevertheless, for specific applications aiming at supporting real-world decisions, for example, evaluating aesthetic impacts of wind power farms, different weather conditions are often included [39, 80]. For this purpose, eye-tracking simulation may be a useful tool to detect changes in contrasts and to test different appearance of the infrastructures to reduce their visibility and thus, their impact on landscape aesthetic values. More generally, eye-tracking simulation may support visual impact assessment in an efficient way. For example, it could improve saliency mapping, which has been recently applied to evaluate visual impacts for wind and solar infrastructure in vineyard landscapes [81]. However, while there is a high correlation between eye-tracking data and saliency maps, it still needs be tested whether eye-tracking simulation provides similar results.

With regard to the composition of the picture (i.e., percentage of sky), our findings corroborate those of Svobodova et al. [55]. This aspect has been considered less often in landscape research, although other studies describe the influence of the characteristics of the horizon line on people's perceptions, e.g., referring to the presence of mountains [29], in urban settings [82], and in relation with fractal characteristics [83, 84]. To reduce the influence of the sky on landscape preferences, our findings suggest that studies need to pay attention to the percentage of sky. Avoiding variations in percentage of sky will therefore produce a better comparability across different landscapes. For photographs taken in landscapes with complex horizon lines (e.g., mountains, differing vegetation height, and tall buildings) this may be more difficult but can be estimated, as done in this study. Using manipulated or computer-generated pictures, the percentage of sky can and should be rigorously controlled.

## Limitations and future research

In general, our findings contribute to a deeper understanding of the influence of clouds in photographs used for eliciting landscape preferences in mountain regions. However, these results may not be transferrable to other landscapes. For example, the importance of sky and

clouds may be differently evaluated in urban environments [43, 83, 85], in flat or hilly land-scapes [86, 87]. Similarly, our results suggest that the influence of clouds was less relevant in pictures with diverse landscape patterns. It is therefore important to include different types of landscapes in future research. Moreover, the manipulations of the sky in some pictures did not completely reproduce natural conditions, as the sky sometimes had little texture and the light-ing of the foreground landscape did not always match the lighting provided by a blue sky. It should therefore paid attention in future studies that the manipulations are not noticeable by the participants to avoid a potential negative influence on the preference scores of the manipu-lated pictures.

While we can assume that our results represented well enough general preferences, as sug-gested by the high correlation of our results with those of two former representative surveys [25, 54], future studies should use a stratified sampling approach, requiring also a higher num-ber of respondents, to assess potential differences across socio-cultural groups. Regarding the composition of the landscape, differences in preferences were found, for example, between lay people and experts [17, 26, 27], and between residents and tourists [25, 26]. Preferences may also depend on the educational level [27], environmental values [30], or the connection of respondents with the landscape [88]. However, it still needs to be verified whether such differ-ences also occur in terms of clouds.

While the above indicated recommendations apply for research focusing on general differ-ences in perceptions across different landscapes, a less studied research field opens up in exam-ining temporary changes, such as seasonal shifts [22, 89, 90] and diurnal changes [43, 85]. For example, different weather and light conditions can moderate preferences for a specific land-scape [43]. The influence of such temporary states of landscapes on cultural ecosystem services is still less explored, despite studies addressing specific 'events,' such as the cherry blossoms [91], wildflower blooms [92], or the occurrence of rainbows [43, 93]. It may therefore not enough to use 'blue-sky' pictures to evaluate the overall aesthetic value of a location but to account for typical weather conditions and seasonal changes. A deeper understanding of the relationships between landscapes and human well-being may also allow decision-making and planning to better scope with global change pressures and integrate cultural ecosystem services into management plans and policies [15].

## Conclusions

Our findings suggest that the presence of clouds in sunny landscape photographs can alter landscape preferences. A high proportion of clouds, in particular in combination with a low proportion of sky, can lead to lower preference scores compared to a sky without clouds. In contrast, a high proportion of sky with single, small clouds can increase preference scores. Our findings therefore have important implications for landscape preference studies using photo-graphs as stimuli, suggesting that such studies need to pay attention to variations in the per-centage of sky and cloud patterns.

However, as we focused on mountain landscapes, further research is needed to verify our findings in other landscapes, such as flat and hilly landscapes or in urban environments. It would be also necessary to apply a stratified sampling approach to assess potential differences in perceptions between respondents with different socio-cultural characteristics. We focused on the appearance of the sky in photographs taken at sunny days, as recommended to ensure comparability of the stated preferences across pictures. Yet, examining variations in appear-ance of the same landscape, for example, under different weather conditions, different times of the day, or different seasonal stages, may improve the understanding of the relationships between landscapes and human well-being.

## Supporting information

**S1 Fig.** Original photographs with clouds (A) and manipulated pictures without clouds (B). Own photographs.
(PDF)

**S2 Fig. Correlation between the results from this study (preference of sample represents) and the general preferences from the studies by Schirpke et al. (2016) and Forer (2020).**
(TIF)

**S3 Fig.** Hotspots in original photographs (A) and manipulated pictures (B) derived from eye-tracking simulation using 3M-VAS. Own photographs.
(PDF)

**S4 Fig. Estimation of landscape composition in terms of distance zones.** Blue = near zone ($< 60$ m), red = middle zone ($> 60$ m– 1.5 km), yellow = far zone ($>1.5$ km), and auxiliary grids to determine the area fraction of distance zones and landscape features. Own photograph.
(TIF)

**S5 Fig. Contribution of visual elements to the overall probability that areas are seen within the first 3–5 seconds.** The yellow circles indicate the hotspots identified in 3M-VAS. Within each hotspot, the importance of each visual element was estimated on a scale of from 0 to 100%. For example, for hotspot no. 3, edges take up 50% of the area, intensity has mostly medium to high values (grey to light grey areas), red-green contrast values are low (dark grey patterns) and blue-yellow contrasts as well as no values for faces (black) are missing. Estimated contributions are 50% for edges, 40% for intensity, 5% for red-green color contrast and 0% for blue-yellow color contrast and faces. Own photograph.
(TIF)

**S6 Fig. Grid with 3x3 cells according to the Rule of Thirds.** This grid was used to determine a spatial shift between pictures, for which the preference scores significantly differed between the original picture with clouds and the manipulated picture without clouds. Own photograph.
(TIF)

**S1 Table. Demographic information of the survey participants (n = 124).** This information was not required to complete the questionnaire, and some participants did not at all or only partly indicate demographic details.
(DOCX)

**S2 Table. List of all variables derived from eye-tracking simulation as proposed by Schirpke et al. [1].** (1) Primary hotspot characteristics (i.e., quantitative and spatial information generated by 3M-VAS) and (2) secondary hotspot characteristics (i.e., variables in relation to the different distance zones as well as natural and artificial features within the hotspots).
(DOCX)

**S3 Table. List of all variables derived from photo content analysis as proposed by Schirpke et al. [1].**
(DOCX)

**S4 Table. Datasets used to calculate landscape metrics.**
(DOCX)

**S5 Table. List of all variables derived from Geographic Information System (GIS)-based analysis as proposed by Schirpke et al. [1].**
(DOCX)

**S6 Table. Predictor variable values used as input for the regression analysis.**
(XLSX)

**S7 Table. Differences in perceptions for 29 photo pairs with clouds (A) and without clouds (B).** Significant differences are formatted in bold (sample size: 124 participants).
(DOCX)

**S8 Table. Mean values and standard deviation (SD) of the variables sky and clouds across picture pairs with decreased (A>B), not changed (A = B), and increased (A<B) preference score.** A: Original picture with clouds, B: manipulated picture without clouds.
(DOCX)

**S9 Table. Spatial shift hotspots due to cloud removal.** Only pictures with significant differences between preference scores of the original and the manipulated picture are included. The spatial shift was determined based on a grid with 9 cells that was overlaid over the pictures (see S6 Fig).
(DOCX)

**S10 Table. Main cloud types in the original pictures.**
(DOCX)

**S1 File.**
(PDF)

## Acknowledgments

We thank all respondents for participating in the survey. The authors are grateful to Kelly Canavan, Global Marketing Development Manager for VAS at 3M Company for allowing the use of 3M-VAS software. We would also like to thank Mr. Wolfgang Gurgiser (meteorologist) for his help in determining the cloud types.

## Author Contributions

**Conceptualization:** Erich Tasser, Alexandros A. Lavdas, Uta Schirpke.

**Data curation:** Erich Tasser, Alexandros A. Lavdas, Uta Schirpke.

**Formal analysis:** Erich Tasser, Alexandros A. Lavdas, Uta Schirpke.

**Methodology:** Erich Tasser, Uta Schirpke.

**Visualization:** Erich Tasser, Uta Schirpke.

**Writing – original draft:** Erich Tasser, Uta Schirpke.

**Writing – review & editing:** Erich Tasser, Alexandros A. Lavdas, Uta Schirpke.

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
