## [Decision Letter · Decision Letter 0]

19 Apr 2023

PONE-D-23-07973Do clouds in landscape photographs influence people’s preferences?PLOS ONE

Dear Dr. Schirpke,

Thank you for submitting your manuscript to PLOS ONE. After careful consideration, we feel that it has merit but does not fully meet PLOS ONE’s publication criteria as it currently stands. Therefore, we invite you to submit a revised version of the manuscript that addresses the points raised during the review process.

We look forward to receiving your revised manuscript.

Kind regards,

Chaohai Shen

Academic Editor

PLOS ONE

Journal Requirements:

2. You indicated that ethical approval was not necessary for your study. We understand that the framework for ethical oversight requirements for studies of this type may differ depending on the setting and we would appreciate some further clarification regarding your research. Could you please provide further details on why your study is exempt from the need for approval and confirmation from your institutional review board or research ethics committee (e.g., in the form of a letter or email correspondence) that ethics review was not necessary for this study? Please include a copy of the correspondence as an ""Other"" file

3. Please amend your current ethics statement to address the following concerns:

a) Did participants provide their written or verbal informed consent to participate in this study?

Additional Editor Comments (if provided):

Dear Authors,

I have received all the required reviewer reports. I agree with the comments from the reviewers and I would like to invite you to revise the manuscript accordingly.

Thank you!

Sincerely,

Reviewers' comments:

Reviewer's Responses to Questions

**Comments to the Author**

1. Is the manuscript technically sound, and do the data support the conclusions?

Reviewer #1: Yes

Reviewer #2: Partly

Reviewer #3: Partly

2. Has the statistical analysis been performed appropriately and rigorously? 

Reviewer #1: I Don't Know

Reviewer #2: Yes

Reviewer #3: I Don't Know

3. Have the authors made all data underlying the findings in their manuscript fully available?

Reviewer #1: No

Reviewer #2: No

Reviewer #3: No

4. Is the manuscript presented in an intelligible fashion and written in standard English?

Reviewer #1: Yes

Reviewer #2: No

Reviewer #3: Yes

5. Review Comments to the Author

Reviewer #1: I'm not able to see where they mention a repository and I don't see their data attached as supplementary, so I answered no to that question above. I may simply have just missed something.

For the authors:

This well-crafted and explained study explores the impacts of clouds in landscape photographs in mountainous settings, using paired original and manipulated photos. The combination of simulated eye-tracking, photo structural analysis and landscape metrics provide a rich assessment of when clouds matter and when they do not. I do not have the statistics skills to assess their quantitative methods, so hope another reviewer is doing that work. But as a landscape researcher I can take important methodological advice from this work.

A few details of the survey are unclear. Could the authors confirm if survey respondents received all 58 images, or a subset of them? That is a large set of photos to review for any one respondent. The time that the survey took suggests the respondents would spend only 5-10 seconds per photo, which seems to be a short time to carry out such cognitive tasks over such a number of them. I also wonder if the survey respondents were incentivized at all? I can’t imagine getting so many people to do a survey of this type without offering academic credit, or a chance to win something. Were the students in any particular program, and how were they invited? Demographics were clearly collected, but are not reported. Such details are generally disclosed, even though the demographics are not the key interest in this survey.

I also enjoyed learning about the 3M-VAS software, but I think it is important to provide more about how the software works. Looking at the webpage it clearly has an AI engine, but can the authors provide a brief explanation of this software (as far as is known, given its commercial status) so that we can assess replicability? Is the program using a similar process as is described elsewhere as ‘salience mapping’? If so, it may be appropriate to connect those literatures. For instance, in the discussion the authors mention the potential value of such methods in understanding the visual impact of wind turbines, and this has been done with salience maps. (e.g., Mohammadi, Mehrnoosh, et al. "A saliency mapping approach to understanding the visual impact of wind and solar infrastructure in amenity landscapes." Impact Assessment and Project Appraisal 41.2 (2023): 154-161.)

A little more detail would also be useful in the section discussing the regression analysis. Can the authors describe exactly what the dependent variable is? Is it A-B?

The graphics are largely effective and well-designed, but Figure 2 needs a little more work. The A and B becomes confusing because there are two uses of these in the figure, for the two parts of the figure and for the cloud and no-cloud image versions. In black and white, the colours for A>B and A=B are identical, and the link to the border of the images is lost. I would suggest bars instead of a pie, extending beside each of the photos, with labels alongside, to completely disambiguate. This would also remove the need for this figure to be seen as two things, as the elements would be entirely integrated.

Language is clear. Only one “und”, and minor issues like “ration” instead of “ratio” in one spot, and “contribute” instead of “contribution” through Table 1.

There is an indication in the cover materials that the authors have made their data fully available, but I’m not able to see where it is attached or pointed to in a repository. My apologies if I simply missed this.

Reviewer #2: Dear Colleagues,

I have read and reviewed your article titled, Do clouds in landscape photographs influence people’s preferences? The research is nicely done and I myself has learned a lot as well. As this is one of my favourite research areas and a discipline, I am familiar with, I would like to make some comments about the manuscript and other aspects. I am grateful for the authors for their attempt in this study which would contribute highly to the landscape and perception related research in the domain.

Some of these comments might be not 100% relevant but I would like to suggest those to improve the structure and the overall quality of your paper.

The title of the manuscript seems a bit too simple. There is only one question as the title. If we incorporate some extra phrase to provide more insights to the manuscript following the question, then I think it would be better. EX:- , Do clouds in landscape photographs influence people’s preferences?; A study in XXXX, or else.

The structure needs a bit more arranging. The titles, Collection of variables, Photo-based survey, Study design can be all made into one topic of "materials and methods". The content in the statistical analysis can be shifted under methods/ results appropriately,

Irrespective of the suggested rearrangement, I will comment based on the existing topics (by the author)

Abstract

Try to be more emphasized on the study and be more focussed. This abstract focusses more on the results which is a good point. But the concluding sentence can be more made useful. Think if some one wants to read your whole article, the abstract is what gets their attention. So try to attract them with the concluding sentence.

Line 14 – 16 – Rewrite the sentence better.

Write the purpose of the study more focussed in the abstract.

Introduction

This is okay, but I would like to have some more insights to the clouds in the intro. Shapes of the clouds and some other information you have discussed in the discussion part seems to be fitting more in the introduction. The clouds and its visible aspects in the landscapes is an interesting part related to the perception research. If you could include those information, then it would be more interesting for the readers.

Line 77 – Sentence should start with a new line.

Photo based survey

Line 94 – Give credits to photoshop

Line 102 – it took about 5-10 minutes ........ – this part is too casual. May be the estimated time to fill the questionnaire?

Line 103 – What is the process of selecting the respondents? A proper sampling technique?

Why 10 point likert scale? Why not an odd number (with a mid point - neutral response)? Why 10 points? Please explain why these were selected for the study.

What is the basis for selecting the photographs? Please explain why?

The general preference of the sample is justified using previous studies. How can you be sure the previous samples reflect general conditions? It is true that high numbers correspond to the normality. Is this comparison to previous studies necessary? Think again.

Ethics statement

This in the middle of the manuscript? Better to include in the body without having a separate section

Line 124 - Adult persons or Adults?

Collection of Variables

Give credits to 3M-VAS software. This information is missing in the manuscript. For the softwares used (manufacturer, country) should be mentioned in parentheses

line 136 – 3 -5 seconds?

photo content analysis – How the structural composition was estimated? Using which software or how? Please explain properly.

We estimated the composition of the landscape in terms of distance zones (i.e., near zone, middle zone, far zone). An illustration of how this was done would be better.

Statistical Analysis

Why backward stepwise linear regression was used? Explain

Results

Line 205 – R2

Line 236 – und?

Line 238 – seconds?

Line 238 – give credits right next to the photographs in the description.

The red and yellow markings in the photographs? What are those?

Each photograph has two analysis shown right? Explain out outcome is this? What these numbers represent. May be I am not quite familiar with the software, I find it difficult to understand. But better if we can include them to increase the readability and for greater exposure.

Where is the conclusion? A paper with no conclusion does not sound right.

The paper is interesting and a great attempt to contribute to the existing literature. Congratulations on the future as a scholar.

Regards!

Reviewer #3: Title

Do clouds in landscape photographs influence people’s preferences?

Summary

The authors present a study examining how clouds, and a host of other independent variables, might affect preference ratings for landscape views. Results indicate differences between images with and without clouds, and according to the proportion of sky depicted. The paper attempts to relate these findings to a cultural ecosystem services framework.

Overall comments

Whilst I endorse the authors’ focus on this overlooked component of landscape evaluation, in its current form the paper does not provide adequate information about the study design or its participants, and reporting of results requires substantial additional information.

The introduction, and materials and methods, should be improved to help the reader understand the visual stimuli being tested and the demographic characteristics of participants. The limited information presented here renders the study impossible to reproduce.

The results are difficult to follow and interpret, and seem to combine empirical findings with visually inferred observations. It is difficult to assess the validity of the statistical approach with the information provided. This section needs careful rewriting to present the results in a way that readers can both understand and suitably interrogate.

The authors must also consider publishing their image sets, analysis scripts, and raw data on the Open Science Framework (or similar platform) such that their findings can be reproduced.

Specific comments

#1 The intentions of the study could be clearer in the abstract. For example at line #13 “Usually, pictures with similar weather and light conditions (i.e., blue sky) are used to reduce the potential influence on landscape preferences” should be rephrased to explain what potential confounders ‘blue sky’ conditions are mitigating.

#2 The introduction would benefit from further information on the different forms clouds can take. The example photographs included in the paper depict various types of cumulus clouds, were these the only cloud formations included across the images used? For example, participant responses to cirrus stratus or altocumulus may have varied substantially compared to cumulus clouds. Moreover, why did the authors choose the cloudscapes included in this study over other formations?

#3 An overview of the methods typically used to assess landscape preferences in photographs would help the reader understand how aesthetic value is usually measured, and therefore how this study's methods might be applicable to other work in this field.

#4 The discussion on study design should include an explanation of how the authors arrived at a suitable sample size. Was a power calculation performed before recruitment? What sample sizes are common in other studies that assess landscape aesthetics?

#5 What were the characteristics of the participants? The authors note that the survey was distributed to undergraduates and their family members, but no information is provided on who actually took part and therefore who their findings relate to. Please include a breakdown of participant demographics, with n’s for each grouping specified.

#6 The authors note that the survey took 5-10 minutes to complete (line #102). This description would benefit from further details. Where did participants complete the study, at home or in a controlled laboratory setting? On what kind of device? How were the images displayed and for how long? Was completion time monitored, and any quality control in place to prevent people from ‘speeding’ through without paying attention? Were participants primed as to the purposes of the study? Virtual eye tracking was used but few details are provided on how and why this method is appropriate, how does it compare to actual eye tracking? On this point, the authors simply refer to “Schirpke et al.” but I suggest that the reader should be presented with these points without having to refer to another study – were the elements of that study followed exactly? How do the methods here deviate?

#7 At line #108 the authors refer to “two former surveys” which the reader assumes are those referenced previously. If this is the case, it should be clarified which studies these are with appropriate referencing and details where necessary. For example, why is it appropriate to compare preference scores from this study to those mentioned?

#8 At line 152 the authors note that “areas hidden by vegetation, buildings, or mountains, were excluded from further analysis.” The reader is curious why it is appropriate for these potential confounding elements to be omitted from the analysis. The authors should provide justification for this approach in the manuscript.

#9 Why it is acceptable to replace missing values with mean values (line #167)? Did this method apply to all IVs? Could the authors explain this in a way such that a reader who is unfamiliar with statistical methods can understand the steps taken.

#10 At line #176 the authors mention tests for normality but do not provide results from these tests until line #206. I suggest clarifying these outcomes here or at least acknowledging that tests were passed and reported later.

#11 To aid interpretation of the results, Figure 2 would be clearer if both A and B images were shown for each example. At present the comparisons being assessed are not clear.

#12 Tabular data underpinning Figure 3 should be displayed. Importantly, what manipulation to the data has been applied to create this figure? Are these mean preference scores across groupings? Are these differences across groups statistically significant? If so, what test has been applied to determine this relationship?

#13 Furthermore, it is not clear how the percentages reported in line #196 have been deduced. Further clarification is required to help these results be interpreted and reproduced.

#14 Table 2 is difficult to interpret in its current form since IVs have been described by their coded descriptors (which the reader can only interpret by cross referencing with Table S1 in supplementary materials). I suggest these are replaced with more easily interpretable labels, for example, “I_HP1..n (%)” could read “Sky within hotspots (%)”. Furthermore, significant coefficients in table 2 might be highlighted in bold to improve readability.

#15 Figure 4 shows some intriguing patterns, but I found it very difficult to follow the descriptions provided in lines #212 to #230. Are the authors referring to results from table 2? If so, it would be useful to report coefficients and p-values in these descriptions so the reader can understand the relationships being described. More importantly, these descriptions appear to refer to images grouped according to whether they do or do not contain clouds. Were two regression models run based on these categories? The IV in the table simply refers to % cloud cover, and clouds Yes/No as a binary variable is not included nor as an interaction term. Greater clarification is required on how the authors can qualify statements such as “Pictures with clouds that had higher preference scores comprised on average more hotspots” and “preferences for landscapes without clouds were often related to the presence of a large top hotspot”. Have these been empirically tested or visually inferred from the data?

#16 At line #220 the authors suggest that “a high proportion of background (EZ4_P1..n) also reduced the negative effects of anthropogenic structures in the foreground (Eart_P1..n)”. How was this dependency tested? No interaction effects between IVs are reported.

#17 Likewise, at line #299 the authors report that “the hotspots in the more attractive image pair are located more towards the sides than in the center.” Was this empirically tested or is this an observation based on one or more photos?

#18 I am unclear on how the 40% threshold for cloud cover noted in line #246 has been derived. The discussion section should not be the first place results are reported.

#19 More importantly, and in relation to comment #2, nimbostratus clouds are mentioned in line #247, and single cumulus clouds mentioned in line #249. This is the first time cloud type has been introduced and suggests substantial heterogeneity in the clouds depicted. More information is required on the composition of these cloud forms in the introduction and materials and methods, how has this been accounted for as a potential confounding factor?

#20 The authors present a quantitative analysis of how clouds might impact preferences for landscape scenes, and attempt to relate this to cultural ecosystem services. However, as hinted at by the phrase ‘cultural’, these values can be highly subjective. It would be valuable to the reader to understand the limitations of this approach, particularly with respect to the fertile literature on subjective vs objective aesthetic assessments.

6. PLOS authors have the option to publish the peer review history of their article (what does this mean?). If published, this will include your full peer review and any attached files.

Reviewer #1: No

Reviewer #2: No

Reviewer #3: No

---

## [Author Response · Author response to Decision Letter 0]

12 May 2023

Reviewer #1:

I'm not able to see where they mention a repository and I don't see their data attached as supplementary, so I answered no to that question above. I may simply have just missed something.

For the authors:

This well-crafted and explained study explores the impacts of clouds in landscape photographs in mountainous settings, using paired original and manipulated photos. The combination of simulated eye-tracking, photo structural analysis and landscape metrics provide a rich assessment of when clouds matter and when they do not. I do not have the statistics skills to assess their quantitative methods, so hope another reviewer is doing that work. But as a landscape researcher I can take important methodological advice from this work.

Response: Thank you for your time and effort to review our manuscript. We appreciate your encouraging evaluation and the valuable suggestions on how to improve the manuscript.

. 

A few details of the survey are unclear. Could the authors confirm if survey respondents received all 58 images, or a subset of them? That is a large set of photos to review for any one respondent. The time that the survey took suggests the respondents would spend only 5-10 seconds per photo, which seems to be a short time to carry out such cognitive tasks over such a number of them. I also wonder if the survey respondents were incentivized at all? I can’t imagine getting so many people to do a survey of this type without offering academic credit, or a chance to win something. Were the students in any particular program, and how were they invited? Demographics were clearly collected, but are not reported. Such details are generally disclosed, even though the demographics are not the key interest in this survey.

Response: The respondents had to evaluate all 58 images. We agree that 5 minutes were too short and now indicate about 10 minutes after recording the time again in a test run. The participants could also pause the survey by saving their responses and continue at a later time. We invited students in landscape ecology and physical geography during lectures on landscape perceptions by providing a link to the questionnaire. The participants received neither a compensation nor other incentives. We added these details and further information on the survey asked by the other reviewers to the methods. We added a short description of the main characteristics of the respondents in the manuscript. We also provide a table with full demographic details in the supplementary material (Table S1).

I also enjoyed learning about the 3M-VAS software, but I think it is important to provide more about how the software works. Looking at the webpage it clearly has an AI engine, but can the authors provide a brief explanation of this software (as far as is known, given its commercial status) so that we can assess replicability? Is the program using a similar process as is described elsewhere as ‘salience mapping’? If so, it may be appropriate to connect those literatures. For instance, in the discussion the authors mention the potential value of such methods in understanding the visual impact of wind turbines, and this has been done with salience maps. (e.g., Mohammadi, Mehrnoosh, et al. "A saliency mapping approach to understanding the visual impact of wind and solar infrastructure in amenity landscapes." Impact Assessment and Project Appraisal 41.2 (2023): 154-161.)

Response: We now added a detailed description of the 3M-VAS software. Thank you for your question about saliency mapping. Indeed, there is a high correlation between eye-tracking outputs obtained in experiments and salience maps, but it is still untested whether the simulation software would also be suitable. We added a short reflection on this in the discussion section.

A little more detail would also be useful in the section discussing the regression analysis. Can the authors describe exactly what the dependent variable is? Is it A-B?

Response: We agree that this was not clear. Indeed, we used the difference in preference score between the original landscape photograph with clouds (A) and the manipulated picture without clouds (B) as dependent variable. We now added this explanation to the manuscript.

The graphics are largely effective and well-designed, but Figure 2 needs a little more work. The A and B becomes confusing because there are two uses of these in the figure, for the two parts of the figure and for the cloud and no-cloud image versions. In black and white, the colours for A>B and A=B are identical, and the link to the border of the images is lost. I would suggest bars instead of a pie, extending beside each of the photos, with labels alongside, to completely disambiguate. This would also remove the need for this figure to be seen as two things, as the elements would be entirely integrated.

Response: Thank you for the suggestion on how to improve figure 2. We revised the figure accordingly.

Language is clear. Only one “und”, and minor issues like “ration” instead of “ratio” in one spot, and “contribute” instead of “contribution” through Table 1.

Response: The manuscript has been checked again for grammar and spelling.

There is an indication in the cover materials that the authors have made their data fully available, but I’m not able to see where it is attached or pointed to in a repository. My apologies if I simply missed this.

Response: Thank you for indicating this issue. Several data were indeed lacking in the supplementary material, which we now remedied.

Reviewer #2: 

Dear Colleagues,

I have read and reviewed your article titled, Do clouds in landscape photographs influence people’s preferences? The research is nicely done and I myself has learned a lot as well. As this is one of my favourite research areas and a discipline, I am familiar with, I would like to make some comments about the manuscript and other aspects. I am grateful for the authors for their attempt in this study which would contribute highly to the landscape and perception related research in the domain.

Some of these comments might be not 100% relevant but I would like to suggest those to improve the structure and the overall quality of your paper.

Response: Thank you for your time and effort to review our manuscript. We appreciate your encouraging evaluation and the valuable suggestions on how to improve the manuscript.

The title of the manuscript seems a bit too simple. There is only one question as the title. If we incorporate some extra phrase to provide more insights to the manuscript following the question, then I think it would be better. EX:- , Do clouds in landscape photographs influence people’s preferences?; A study in XXXX, or else.

Response: We added a short extra phrase. The title now is: Assessing landscape aesthetic values: Do clouds in photographs influence people’s preferences?

The structure needs a bit more arranging. The titles, Collection of variables, Photo-based survey, Study design can be all made into one topic of "materials and methods". The content in the statistical analysis can be shifted under methods/ results appropriately,

Irrespective of the suggested rearrangement, I will comment based on the existing topics (by the author)

Response: The sections from ‘Study design’ to ‘Statistical analyses’ are all included in the ‘materials and methods’ section. Unfortunately, the journal does not use numbering, which would be clearer. We now changes text size to differentiate more clearly main sections and subsections.

Abstract

Try to be more emphasized on the study and be more focussed. This abstract focusses more on the results which is a good point. But the concluding sentence can be more made useful. Think if some one wants to read your whole article, the abstract is what gets their attention. So try to attract them with the concluding sentence.

Response: We revised the abstract, particularly the concluding sentences, to better focus on the study.

Line 14 – 16 – Rewrite the sentence better.

Write the purpose of the study more focussed in the abstract.

Response: We rephrased the sentence to indicate more clearly the issues that are related to the weather and light conditions in the pictures and to better focus the purpose of the study.

Introduction

This is okay, but I would like to have some more insights to the clouds in the intro. Shapes of the clouds and some other information you have discussed in the discussion part seems to be fitting more in the introduction. The clouds and its visible aspects in the landscapes is an interesting part related to the perception research. If you could include those information, then it would be more interesting for the readers.

Response: We agree that the information on the clouds would be better placed in the introduction. We therefore moved a part of it from the discussion to the introduction section, which we also integrated with some further information.

Line 77 – Sentence should start with a new line.

Response: This is still part of the caption. It is the figure legend and was formatted according to the author guidelines. To clearer distinguish the main text from figure and table captions, we slightly reduced the text size of the captions.

Photo based survey

Line 94 – Give credits to photoshop

Response: Done.

Line 102 – it took about 5-10 minutes ........ – this part is too casual. May be the estimated time to fill the questionnaire?

Response: Thank you for your suggestion. We revised the sentence accordingly.

Line 103 – What is the process of selecting the respondents? A proper sampling technique?

Response: We used a convenience sampling approach. We are aware of the limited representativeness of the sample. Therefore, we compared our results to those of the two former surveys, which used a stratified sampling approach (see also below).

Why 10 point likert scale? Why not an odd number (with a mid point - neutral response)? Why 10 points? Please explain why these were selected for the study.

Response: There is no consensus among landscape preference studies on the most appropriate rating scale. We used an even scale, because we wanted the respondents to decide whether their preference for an image is more positive or negative. Moreover, response scales from 7 to 10 are most reliable, and respondents prefer the 10-point scale the most (Preston and Colman, 2000). We added this information in the manuscript.

What is the basis for selecting the photographs? Please explain why?

Response: We used landscape photographs that were used in two former surveys to gather landscape preferences in the Central Alps. The photographs depicted different mountain landscapes that were typical for this area. Despite a blue sky, the photographs contained variations in cloud patterns, ranging from almost no clouds to a sky largely covered by clouds, but no threatening cloud atmosphere. From this pool of 49 landscape photographs, we selected 29 photographs with varying cloud cover to not overload the respondents with a too high number of pictures. We revised the description to explain in greater detail the selection of the photographs for this study.

The general preference of the sample is justified using previous studies. How can you be sure the previous samples reflect general conditions? It is true that high numbers correspond to the normality. Is this comparison to previous studies necessary? Think again.

Response: We used the same pictures in our study as the two former survey, which gathered landscape preferences in a comparable way between 2016 and 2019 in five different study areas in the Central Alps. Both studies used a stratified sampling approach based on the demographic distribution in the study areas, i.e., considering key socio-demographic characteristics such as gender, age, origin (residents, tourists), place of living (village, city), and native language (German, Italian). The two surveys, comprising 967 respondents [30] and 384 respondents [35], can therefore be considered to be representative for the general public of the Central Alps. Moreover, in both surveys, differences between socio-demographic groups in preference scores were generally very small, although some significant differences occurred, mostly between people of different language, age, and origin [30]. We revised the description of the methods by adding these details.

Ethics statement

This in the middle of the manuscript? Better to include in the body without having a separate section

Response: During submission, the journal manager asked us to include it as a separate section.

Line 124 - Adult persons or Adults?

Response: We changed to ‘adults’.

Collection of Variables

Give credits to 3M-VAS software. This information is missing in the manuscript. For the softwares used (manufacturer, country) should be mentioned in parentheses

Response: Added.

line 136 – 3 -5 seconds?

Response: Corrected.

photo content analysis – How the structural composition was estimated? Using which software or how? Please explain properly.

Response: We estimated the structural composition of the pictures by visual analysis. To systematically analyze all pictures, we applied the Rule of Thirds and divided the pictures into thirds both horizontally and vertically [56,57]. We then further subdivided the cells to obtain a grid with 3x9 cells for a finer subdivision of the pictures. This grid was overlaid over each picture to determine the area of landscape features and the composition of the landscape in terms of distance zone s (i.e., near zone, middle zone, far zone). All variables were estimated by one researcher, counting the coverage (%) of the features in each cell and summarizing the percentages across all cells. We added this explanation to the methods section. In addition, we created a new figure to illustrate the approach.

We estimated the composition of the landscape in terms of distance zones (i.e., near zone, middle zone, far zone). An illustration of how this was done would be better.

Response: We revised the description to clearly indicate how this was done and created a new figure to illustrate the approach. See also previous comment.

Statistical Analysis

Why backward stepwise linear regression was used? Explain

Response: The backward stepwise approach starts from a full model with all predictors and successively reduces them to obtain a model that best explains the data, while solving problems of multicollinearity and overfitting. We added this explanation to the methods section.

Results

Line 205 – R2

Response: Corrected.

Line 236 – und?

Response: Corrected.

Line 238 – seconds?

Response: Yes, corrected.

Line 238 – give credits right next to the photographs in the description.

Response: Done.

The red and yellow markings in the photographs? What are those?

Each photograph has two analysis shown right? Explain out outcome is this? What these numbers represent. May be I am not quite familiar with the software, I find it difficult to understand. But better if we can include them to increase the readability and for greater exposure.

Response: The red and yellow markings indicate the hotspot area. Hotspots with a probability >=80% are indicated in red, while those with a probability <80 are indicated in yellow. We added an explanation to the figure caption. Moreover, we added a figure to the supplementary material (Figure Sx), explaining the outputs of the eye-tracking software and we refer to this figure in the methods section.

Where is the conclusion? A paper with no conclusion does not sound right.

Response: We added a short conclusion section.

The paper is interesting and a great attempt to contribute to the existing literature. Congratulations on the future as a scholar.

Regards!

Reviewer #3: 

Title

Do clouds in landscape photographs influence people’s preferences?

Summary

The authors present a study examining how clouds, and a host of other independent variables, might affect preference ratings for landscape views. Results indicate differences between images with and without clouds, and according to the proportion of sky depicted. The paper attempts to relate these findings to a cultural ecosystem services framework.

Overall comments

Whilst I endorse the authors’ focus on this overlooked component of landscape evaluation, in its current form the paper does not provide adequate information about the study design or its participants, and reporting of results requires substantial additional information.

Response: Thank you for your time and effort to review our manuscript. We appreciate your evaluation and the valuable suggestions on how to improve the manuscript. See also comments below.

The introduction, and materials and methods, should be improved to help the reader understand the visual stimuli being tested and the demographic characteristics of participants. The limited information presented here renders the study impossible to reproduce.

Response: We now integrated further information into the introduction to better frame our study and to provide more background information. We also added more details in the methods section to assure replicability of the study. See also comments below.

The results are difficult to follow and interpret, and seem to combine empirical findings with visually inferred observations. It is difficult to assess the validity of the statistical approach with the information provided. This section needs careful rewriting to present the results in a way that readers can both understand and suitably interrogate.

Response: In addition to having added more details in the methods section, we also revised the description of the results to improve clarity and readability. See also comments below.

The authors must also consider publishing their image sets, analysis scripts, and raw data on the Open Science Framework (or similar platform) such that their findings can be reproduced.

Response: We added all images and raw data to the supplementary material.

Specific comments

#1 The intentions of the study could be clearer in the abstract. For example at line #13 “Usually, pictures with similar weather and light conditions (i.e., blue sky) are used to reduce the potential influence on landscape preferences” should be rephrased to explain what potential confounders ‘blue sky’ conditions are mitigating.

Response: We rephrased several parts of the abstract to indicate more clearly the issues that are related to the weather and light conditions in the pictures.

#2 The introduction would benefit from further information on the different forms clouds can take. The example photographs included in the paper depict various types of cumulus clouds, were these the only cloud formations included across the images used? For example, participant responses to cirrus stratus or altocumulus may have varied substantially compared to cumulus clouds. Moreover, why did the authors choose the cloudscapes included in this study over other formations?

Response: We agree that information on the clouds needs to be introduced in the introduction. We therefore moved the text reflecting on perceptions of clouds from the discussion section to the introduction section, also adding some further information. Furthermore, we revised the methods and results section to provide details on clouds for all photographs. As our aim was to assess the influence of clouds in photographs taken at sunny days (as recommended for studies using photographs to gather landscape preferences), only certain cloud types occurred.

#3 An overview of the methods typically used to assess landscape preferences in photographs would help the reader understand how aesthetic value is usually measured, and therefore how this study's methods might be applicable to other work in this field.

Response: We added a short overview in the introduction session on how landscape preferences are usually assessed. We introduce expert-based and perception-based approaches. We explain how landscape preferences are usually assessed, i.e., photo-based questionnaires, which ask the participants to evaluate different photographs using rating scales or applying discrete choice experiments.

#4 The discussion on study design should include an explanation of how the authors arrived at a suitable sample size. Was a power calculation performed before recruitment? What sample sizes are common in other studies that assess landscape aesthetics?

Response: To ensure a minimum sample size for statistical analysis , i.e., to set up a significant regression model, we determined the minimum sample size through statistical power analysis. With a determination coefficient of R² = 0.7, a statistical power of 0.9 and a significance level of α = 0.01, we needed a sample size of at least 68 valid responses. The sample size in other landscape preference studies is highly variable, depending whether they are also analysing differences among socio-demographic groups. Nevertheless, sample sizes between 100 and 200 respondents are very common. We now added an explanation regarding the sample size to the methods section.

#5 What were the characteristics of the participants? The authors note that the survey was distributed to undergraduates and their family members, but no information is provided on who actually took part and therefore who their findings relate to. Please include a breakdown of participant demographics, with n’s for each grouping specified.

Response: We added a short description of the main characteristics of the respondents in the manuscript. We also provide a table with full details in the supplementary material (Table S1).

#6 The authors note that the survey took 5-10 minutes to complete (line #102). This description would benefit from further details. Where did participants complete the study, at home or in a controlled laboratory setting? On what kind of device? How were the images displayed and for how long? Was completion time monitored, and any quality control in place to prevent people from ‘speeding’ through without paying attention? Were participants primed as to the purposes of the study? 

Response: The respondents completed the questionnaire in an unsupervised setting, using a digital device. There were no time restrictions in viewing and rating the photographs. The participants could also pause the survey by saving their responses and continue at a later time. In the first section of the questionnaire, participants were informed about the scope of the survey and that the participation in the survey was voluntary and anonymous. The participants received neither a compensation nor other incentives. The completion time was not monitored, but participants who were not interested in rating the photographs did not complete the questionnaire. Indeed, 176 people opened the questionnaire, but only 124 also completed it. We added these details to section on the photo-based survey.

Virtual eye tracking was used but few details are provided on how and why this method is appropriate, how does it compare to actual eye tracking? On this point, the authors simply refer to “Schirpke et al.” but I suggest that the reader should be presented with these points without having to refer to another study – were the elements of that study followed exactly? How do the methods here deviate?

Response: 3M-WAS uses an artificial intelligence (AI) algorithm that is based on 30 years of lab-based eye-tracking research to simulate human pre-attentive processing in vision. Based on five visual elements that are recognized to attract human attention (edges, intensity, red-green color contrast, blue-yellow color contrast, and faces), the algorithm predicts what people would be focusing their eyes on in the initial 3-5 seconds of looking at an image, before the conscious brain can react. 3M-VAS claims to be as reliable and effective as eye tracking hardware, predicting pre-attentive vision with around 92% accuracy. The software also overcomes issues related to repeatability of real eye tracking experiments and allows direct comparison of the outputs.

We added this information to the manuscript. We also revised the description of the analysis by adding further details, e.g., the processing of the images, the analysis of the hotspots. This analysis did not deviate from the study done by Schirpke et al. (2022),

#7 At line #108 the authors refer to “two former surveys” which the reader assumes are those referenced previously. If this is the case, it should be clarified which studies these are with appropriate referencing and details where necessary. For example, why is it appropriate to compare preference scores from this study to those mentioned?

Response: We used the same pictures in our study as the two former survey, which gathered landscape preferences in a comparable way between 2016 and 2019 in five different study areas in the Central Alps. Both studies used a stratified sampling approach based on the demographic distribution in the study areas, i.e., considering key socio-demographic characteristics such as gender, age, origin (residents, tourists), place of living (village, city), and native language (German, Italian). The two surveys, comprising 967 respondents [30] and 384 respondents [35], can therefore be considered to be representative for the general public of the Central Alps. Moreover, in both surveys, differences between socio-demographic groups in preference scores were generally very small, although some significant differences occurred, mostly between people of different language, age, and origin [30]. We revised the description of the methods by adding these details and pointing out more clearly that we used the same pictures as these two surveys.

#8 At line 152 the authors note that “areas hidden by vegetation, buildings, or mountains, were excluded from further analysis.” The reader is curious why it is appropriate for these potential confounding elements to be omitted from the analysis. The authors should provide justification for this approach in the manuscript.

Response: To analyse the landscape seen on the photographs in terms of their composition and configuration through landscape metrics, the maps needed to match the photo content. Thus, we had to remove areas from the map that cannot be seen on the photograph. We revised to description of this paragraph to clearer describe our analysis approach.

#9 Why it is acceptable to replace missing values with mean values (line #167)? Did this method apply to all IVs? Could the authors explain this in a way such that a reader who is unfamiliar with statistical methods can understand the steps taken.

Response: To include all cases (i.e., photographs) in the analysis, missing values are usually replaced by mean values. Mean values are neutral and do not alter the results. However, we noticed that our variables did not include missing values. We therefore removed the sentence to avoid misleading statements.

#10 At line #176 the authors mention tests for normality but do not provide results from these tests until line #206. I suggest clarifying these outcomes here or at least acknowledging that tests were passed and reported later.

Response: We agree that the results from these tests are better placed together with the description of the methods. We now moved them from the results section to the end of the methods section.

#11 To aid interpretation of the results, Figure 2 would be clearer if both A and B images were shown for each example. At present the comparisons being assessed are not clear.

Response: We revised Figure 2 following the suggestion of Reviewer 1. As including both images would have reduced the image size and visibility of the differences, we did not include them here, but we all picture pairs as well as the results of the hotspot analysis have been included in the supplementary material.

#12 Tabular data underpinning Figure 3 should be displayed. Importantly, what manipulation to the data has been applied to create this figure? Are these mean preference scores across groupings? Are these differences across groups statistically significant? If so, what test has been applied to determine this relationship?

Response: We apologize that an explanation was lacking. Yes, these values are mean preference scores across groupings. We now added the following explanation: The values represent mean values of the variables sky and clouds across all picture pairs with decreased (A>B), not changed (A=B), and increased preference score (A<B). We also added tabular data of this figure to the supplementary material (Table S7). Moreover, we tested statistically significant differences using the least significant difference (LSD) post hoc test. We added this in the methods section and in the Table caption.

#13 Furthermore, it is not clear how the percentages reported in line #196 have been deduced. Further clarification is required to help these results be interpreted and reproduced.

Response: The values represent mean values of the variables sky and clouds across all picture pairs without difference in preference score (A=B). We revised this paragraph by correcting the percentages as reported in Figure 3, adding also the percentages for the decreased and increased preference scores, as well as by adding explanations what these represent. See also previous comment.

#14 Table 2 is difficult to interpret in its current form since IVs have been described by their coded descriptors (which the reader can only interpret by cross referencing with Table S1 in supplementary materials). I suggest these are replaced with more easily interpretable labels, for example, “I_HP1..n (%)” could read “Sky within hotspots (%)”. Furthermore, significant coefficients in table 2 might be highlighted in bold to improve readability.

Response: We agree that the codes are difficult to follow. We therefore replaced them with more easily interpretable labels. A description of the selected variables is also provided in Table 1 in the manuscript, while Table S1 contains also all variables that have been excluded during the regression analysis. We decided to not highlight significant coefficients in Table 2, as all significant at a significance level of p<0.1; and only two coefficients have a significance level < p<0.05.

#15 Figure 4 shows some intriguing patterns, but I found it very difficult to follow the descriptions provided in lines #212 to #230. Are the authors referring to results from table 2? If so, it would be useful to report coefficients and p-values in these descriptions so the reader can understand the relationships being described. More importantly, these descriptions appear to refer to images grouped according to whether they do or do not contain clouds. Were two regression models run based on these categories? The IV in the table simply refers to % cloud cover, and clouds Yes/No as a binary variable is not included nor as an interaction term. Greater clarification is required on how the authors can qualify statements such as “Pictures with clouds that had higher preference scores comprised on average more hotspots” and “preferences for landscapes without clouds were often related to the presence of a large top hotspot”. Have these been empirically tested or visually inferred from the data?

Response: We revised the text to make the reference to table 2 more evident, but we decided to not include coefficients and p-values in the main text, as these would reduce readability and are redundant to the information provided in Table 2. In the text, we refer to differences between picture pairs (i.e., original picture with clouds, and modified picture without clouds). We therefore did not need to include a binary variable regarding the clouds. We ran only one regression model using the difference between picture pairs as dependent variable. We now made this clearer in the methods section as well as in the caption of Table 2. 

#16 At line #220 the authors suggest that “a high proportion of background (EZ4_P1..n) also reduced the negative effects of anthropogenic structures in the foreground (Eart_P1..n)”. How was this dependency tested? No interaction effects between IVs are reported.

Response: We did not test interactions between variables. We revised the text to avoid misleading statements.

#17 Likewise, at line #299 the authors report that “the hotspots in the more attractive image pair are located more towards the sides than in the center.” Was this empirically tested or is this an observation based on one or more photos?

Response: This has been empirically tested for all photos, analyzing the pictures according to the Rule of Thirds. We now describe this in the methods section, and we added the results to the supplementary material (Table S8).

#18 I am unclear on how the 40% threshold for cloud cover noted in line #246 has been derived. The discussion section should not be the first place results are reported.

Response: This has been derived from Figure 3. We now report the exact mean values also in results section in the text and revised the discussion section.

#19 More importantly, and in relation to comment #2, nimbostratus clouds are mentioned in line #247, and single cumulus clouds mentioned in line #249. This is the first time cloud type has been introduced and suggests substantial heterogeneity in the clouds depicted. More information is required on the composition of these cloud forms in the introduction and materials and methods, how has this been accounted for as a potential confounding factor?

Response: We moved the text reflecting on perceptions of clouds from the discussion section to the introduction section, also adding some further information. Furthermore, we revised the methods and results section to provide details on clouds for all photographs. We also added the supplementary Table S9, specifying the main cloud types that are present in the original photos.

#20 The authors present a quantitative analysis of how clouds might impact preferences for landscape scenes, and attempt to relate this to cultural ecosystem services. However, as hinted at by the phrase ‘cultural’, these values can be highly subjective. It would be valuable to the reader to understand the limitations of this approach, particularly with respect to the fertile literature on subjective vs objective aesthetic assessments.

Response: People’s perceptions are of course highly subjective and landscape preferences can be very heterogeneous at the individual level. Nevertheless, perception-based approaches are considered to provide reliable results for the general public. We added more information in the introduction about the different approaches, with a focus on perception-based approaches. We also indicate potential differences between different socio-cultural groups. In the discussion section, we inserted a subheading ‘Limitations and future research’ to indicate more clearly the different subsections of the discussion. In this subsection, we also deepened the discussion about the subjectivity of landscape preferences.

---

## [Decision Letter · Decision Letter 1]

18 Jun 2023

PONE-D-23-07973R1Assessing landscape aesthetic values: Do clouds in photographs influence people’s preferences?PLOS ONE

Dear Dr. Schirpke,

Thank you for submitting your manuscript to PLOS ONE. After careful consideration, we feel that it has merit but does not fully meet PLOS ONE’s publication criteria as it currently stands. Therefore, we invite you to submit a revised version of the manuscript that addresses the points raised during the review process.

We look forward to receiving your revised manuscript.

Kind regards,

Chaohai Shen

Academic Editor

PLOS ONE

Journal Requirements:

Reviewers' comments:

Reviewer's Responses to Questions

**Comments to the Author**

1. If the authors have adequately addressed your comments raised in a previous round of review and you feel that this manuscript is now acceptable for publication, you may indicate that here to bypass the “Comments to the Author” section, enter your conflict of interest statement in the “Confidential to Editor” section, and submit your "Accept" recommendation.

Reviewer #1: (No Response)

Reviewer #2: All comments have been addressed

Reviewer #3: (No Response)

2. Is the manuscript technically sound, and do the data support the conclusions?

Reviewer #1: Yes

Reviewer #2: Yes

Reviewer #3: Yes

3. Has the statistical analysis been performed appropriately and rigorously? 

Reviewer #1: I Don't Know

Reviewer #2: Yes

Reviewer #3: Yes

4. Have the authors made all data underlying the findings in their manuscript fully available?

Reviewer #1: Yes

Reviewer #2: Yes

Reviewer #3: Yes

5. Is the manuscript presented in an intelligible fashion and written in standard English?

Reviewer #1: Yes

Reviewer #2: Yes

Reviewer #3: Yes

6. Review Comments to the Author

Reviewer #1: This paper has been improved with revision. I have only a few remaining comments to further strengthen or clarify the work.

139 I acknowledge that there is no set scale for assessing landscape preferences, and this may be an issue of translation, but I am finding least preferred and most preferred somewhat nonsensical for photos explored individually. Least and most are comparative terms. Could the authors check the best translation, if formal equivalence exists, as I’m sure this must have made more sense in German.

I’m finding the ‘intensity of visual elements’ variable a bit tricky to conceptualize, and indeed quite a few of the eye-tracking variables are a bit hard to conceptualize. Is it density or concentration or diversity? The Table 1 isn’t always helpful, given that the units are mostly described as percentages but the descriptions include things like areas and contributions to probability. 235-238 is not easy to understand and I think these steps should be understandable without looking at the Schirpke reference cited.

Some of the results could be clearer if some additional terms could be added, for instance in line 311 add “with clouds” after “lower”, or in line 323, add “in the” before “pair” (if I’m interpreting correctly). Additionally 325 add “with cloud removal after “photos” at the end of that line.

At the end of 345 you indicate a high intensity of visual elements was associated with pictures without clouds. I can’t seem to find the supplemental data reporting this. I see the difference score regression. Maybe include the relevant stats in the body and clearly note where reported where those models can be found.

Discussion starts with reporting on a hypothesis that I don’t remember being declared earlier.

Small things but still worth mentioning.

80-81 This sentence seems tautological, maybe drop the first part

89-93 Given the paragraph that comes before it seems unclear, but somewhat likely that clouds will alter preferences. This paragraph doesn’t seem to acknowledge the one before. It could also be done by noting that it is not just “attracting attention” that affects preferences but the emotional impact and mood that clouds can bring to an image.

143 use “assessed” rather than obtained

156 Is the estimated time to complete the researchers’ estimation or based on actual completion?

402 using “were” reads as if you found it rather than Svobodova. Maybe “have been”?

433 and elsewhere Using “percentage sky” reads overly technical for discussion. Percentage “of” sky

Reviewer #2: I appreciate the effort you have put in revising the manuscript answering all the comments from authors.

I only have one small doubt to clarify,

Starting from line 138 - You have mentioned that ....... to decide whether their preference is positive or negative

But you have included the likert scale as 1 - least preferred (which is the lowest level of preference and obviously not a negative response). Please explain this incase I misunderstood!

Line 209 - 3M-WAS

Line 213 - ...... their eyes on in the initial..... ???

Other than that all the comments are addressed. Great work.

Reviewer #3: I commend the authors on their revised manuscript. They have addressed all reviewer comments and the resulting paper is much improved and offers a valuable contribution to the field.

I have just one comment I think it would be worth addressing before recommending publication:

The manipulated images are, in some cases, very obviously edited. The blue sky has little texture, and the lighting of the foreground landscape does not match that provided by a typical blue sky. It would be valuable to mention this limitation, and its possible effects on ratings; could participant ratings have reflected (at least partly) negative reactions to Photoshopped scenes?

7. PLOS authors have the option to publish the peer review history of their article (what does this mean?). If published, this will include your full peer review and any attached files.

Reviewer #1: No

Reviewer #2: No

Reviewer #3: No

---

## [Author Response · Author response to Decision Letter 1]

26 Jun 2023

Reviewer #1: 

This paper has been improved with revision. I have only a few remaining comments to further strengthen or clarify the work.

Response: Thank you for your time and effort in re-reviewing our manuscript and your positive feedback. We have also now addressed all additional comments.

139 I acknowledge that there is no set scale for assessing landscape preferences, and this may be an issue of translation, but I am finding least preferred and most preferred somewhat nonsensical for photos explored individually. Least and most are comparative terms. Could the authors check the best translation, if formal equivalence exists, as I’m sure this must have made more sense in German.

Response: Thank you for pointing this out. Indeed, there was no comparison among the pictures, but we asked the people to indicate how much they liked the pictures on a scale from 1 = ‘I don’t like it at all’ to 10 = ‘I like it very much’. We now corrected the description in the manuscript.

I’m finding the ‘intensity of visual elements’ variable a bit tricky to conceptualize, and indeed quite a few of the eye-tracking variables are a bit hard to conceptualize. Is it density or concentration or diversity? The Table 1 isn’t always helpful, given that the units are mostly described as percentages but the descriptions include things like areas and contributions to probability.

Response: 3M-VAS describes “intensity” with “luminance contrast, brightness, black/white contrast”. We added this explanation to the description in Table 1. The variables are expressed in %, as it indicates the overall contribution to the probability of looking at a hotspot. Accordingly, we refined the description of these variables in Table 1.

235-238 is not easy to understand and I think these steps should be understandable without looking at the Schirpke reference cited.

Response: We revised the description of the methodological steps by adding all details. We also added an additional figure to the supplementary material (new Fig. S5), which illustrates the estimation of the contribution of the visual elements.

Some of the results could be clearer if some additional terms could be added, for instance in line 311 add “with clouds” after “lower”, or in line 323, add “in the” before “pair” (if I’m interpreting correctly). Additionally 325 add “with cloud removal after “photos” at the end of that line.

At the end of 345 you indicate a high intensity of visual elements was associated with pictures without clouds. I can’t seem to find the supplemental data reporting this. I see the difference score regression. Maybe include the relevant stats in the body and clearly note where reported where those models can be found.

Response: We added the suggested terms or rephrased the sentences to improve clarity. The sentence in line 345 still referred to Table 2, where the results of the regression analysis are reported. We now added the cross-reference to this table to clearly indicate the underlying data. Moreover, we added another explanation in the table caption to clearly indicate the meaning of the coefficients; i.e., a negative regression coefficient B indicates that this variable has a positive effect on the preference score of photos with clouds.

Discussion starts with reporting on a hypothesis that I don’t remember being declared earlier.

Response: We rephrased the sentence to match our hypothesis presented at the end of the introduction.

Small things but still worth mentioning.

80-81 This sentence seems tautological, maybe drop the first part

Response: We removed the first part of the sentence and slightly rephrased the second part.

89-93 Given the paragraph that comes before it seems unclear, but somewhat likely that clouds will alter preferences. This paragraph doesn’t seem to acknowledge the one before. It could also be done by noting that it is not just “attracting attention” that affects preferences but the emotional impact and mood that clouds can bring to an image.

Response: Thank you for this suggestion. We now also indicate that preferences are influence by emotions and moods.

143 use “assessed” rather than obtained

Response: Done.

156 Is the estimated time to complete the researchers’ estimation or based on actual completion?

Response: We could not record the time of the participants to complete the questionnaire. However, we measured how long it took us to fill it out. We rephrased the sentence to clarify that we estimated the time.

402 using “were” reads as if you found it rather than Svobodova. Maybe “have been”?

Response: Done.

433 and elsewhere Using “percentage sky” reads overly technical for discussion. Percentage “of” sky

Response: We changed to “percentage of sky” throughout the manuscript.

Reviewer #2: 

I appreciate the effort you have put in revising the manuscript answering all the comments from authors.

I only have one small doubt to clarify,

Response: Thank you for your time and effort in re-reviewing our manuscript and your positive feedback. We have also now addressed all additional comments.

Starting from line 138 - You have mentioned that ....... to decide whether their preference is positive or negative. But you have included the likert scale as 1 - least preferred (which is the lowest level of preference and obviously not a negative response). Please explain this incase I misunderstood!

Response: Indeed, the description of the rating scale was wrong, as we asked the people to indicate how much they liked the pictures on a scale from 1 = ‘I don’t like it at all’ to 10 = ‘I like it very much’. In this case, values >= 5 can be interpreted as positive and below 5 as negative. We now corrected the description of the rating scale in the manuscript.

Line 209 - 3M-WAS

Response: Corrected.

Line 213 - ...... their eyes on in the initial..... ???

Response: We revised the sentence to clarify the meaning.

Other than that all the comments are addressed. Great work.

Response: Thank you.

Reviewer #3: 

I commend the authors on their revised manuscript. They have addressed all reviewer comments and the resulting paper is much improved and offers a valuable contribution to the field.

I have just one comment I think it would be worth addressing before recommending publication:

Response: Thank you for your time and effort in re-reviewing our manuscript and your positive feedback. We have also now addressed all additional comments.

The manipulated images are, in some cases, very obviously edited. The blue sky has little texture, and the lighting of the foreground landscape does not match that provided by a typical blue sky. It would be valuable to mention this limitation, and its possible effects on ratings; could participant ratings have reflected (at least partly) negative reactions to Photoshopped scenes?

Response: We agree that this could have been partly influenced the preferences scores. We now added this limitation to the discussion section.

---

## [Editor Report · Decision Letter 2]

29 Jun 2023

Assessing landscape aesthetic values: Do clouds in photographs influence people’s preferences?

PONE-D-23-07973R2

Dear Dr. Schirpke,

We’re pleased to inform you that your manuscript has been judged scientifically suitable for publication and will be formally accepted for publication once it meets all outstanding technical requirements.

Kind regards,

Chaohai Shen

Academic Editor

PLOS ONE
---

## [Editor Report · Acceptance letter]

19 Jul 2023

PONE-D-23-07973R2 

Assessing landscape aesthetic values: Do clouds in photographs influence people’s preferences? 

Dear Dr. Schirpke:

I'm pleased to inform you that your manuscript has been deemed suitable for publication in PLOS ONE. Congratulations! Your manuscript is now with our production department. 

Kind regards, 

on behalf of

Dr. Chaohai Shen 

Academic Editor

PLOS ONE